# Image Functions in Neural Networks: A Perspective on Generalization

## Abstract

In this work, we show that training with SGD on ReLU neural networks gives rise to a natural set of functions for each image that are not perfectly correlated until later in training. Furthermore, we show experimentally that the intersection of paths for different images also changes during the course of training. We hypothesize that this lack of correlation and changing intersection may be a factor in explaining generalization, because it encourages the model to use different features at different times, and pass the same image through different functions during training. This may improve generalization in two ways. 1) By encouraging the model to learn the same image in different ways, and learn different commonalities between images, comparable to model ensembling. 2) By improving algorithmic stability, as for a particular feature, the model is not always reliant on the same set of images, so the removal of an image may not adversely affect the loss.

## 1    Introduction

Determining why neural networks generalize remains an interesting open problem. Training often succeeds even without the use of explicit regularizers like Dropout Srivastava et al. (2014), and even when the model is trained all the way to 100% training accuracy, or in the presence of over-parametrization Zhang et al. (2021). Existing bounds using capacity control Neyshabur et al. (2019), e.g. Rademacher complexity or VC dimension are difficult to analyze when there is zero label noise and zero empirical risk Belkin et al. (2018). Several interesting approaches have been explored to explain the generalization phenomenon. These include approaches based on applying algorithmic stability Bousquet & Elisseeff (2000) to SGD Hardt et al. (2016) Kuzborskij & Lampert (2018) , PAC Bayes based bounds Dziugaite & Roy (2017) approaches exploring properties of neural network functions like elasticity He & Su (2019). It has also been hypothesized that neural networks have a spectral bias towards low frequency functions Rahaman et al. (2019) . While discovering neural networks generalize on small datasets, Olson et al. (2018) introduced an algorithm to decompose a neural network into a set of uncorrelated trees, possibly explaining their ability to perform variance reduction.

For many machine learning models, boosting Schapire (2003) and bagging Breiman (1996) are employed to improve generalization performance by either averaging over different trained models, or resampling the dataset so it contains different distributions of data points. The underlying intuition is that different models are unlikely to make the same mistake. One can also train models on different subsets of features and take an average to reduce variance. Additionally, one promising line of investigation surrounding neural network generalization is that of algorithmic stability, as analyzed by Bousquet & Elisseeff (2000) and Hardt et al. (2016) for the case of SGD.

In this work, we consider ReLU networks trained using SGD on image data. We show that training with SGD on ReLU neural networks gives rise to a natural set of functions for each image that are not perfectly correlated until later in training. Furthermore, we show experimentally that the intersection of paths for different images also changes during the course of training. We hypothesize that this lack of correlation and changing intersection may be a factor in explaining generalization, because it encourages the model to use different features at different times, and pass the same image through different functions during training. This may improve generalization in two ways. 1) By

encouraging the model to learn the same image in different ways, and learn different commonalities between images, comparable to model ensembling.

*In particular, at time $t$, an image has access to a **subset** of features from image functions $f_j^t$, and this subset changes over time. Hence, $f_i^t$ may not be too correlated with the end-to-end function $f_j^k$, $k \leq t$ on the training set, allowing for generalization.*

2) By improving algorithmic stability, as for a particular feature, the model is not always reliant on the same set of images, so the removal of an image may not adversely affect the loss.

## 2 PRIOR WORK

Keskar et al. (2019) investigate the effect of sharpness and flatness of minima on generalization, and find that flat minima generalize better. Dinh et al. (2017) find that if the network is re-parametrized, sharp minima can also generalize well. Smith & Le (2018) give a bayesian perspective on generalization and SGD. Dziugaite & Roy (2017) reveal a way to compute non-vacuous generalization bounds using PAC-Bayes He & Su (2019) study the local elasticity of neural networks, where the prediction at $x$ is not significantly perturbed if a gradient update is performed on another dissimilar point $x'$. Deng et al. (2021) use local elasticity to compute a generalization bound. Poggio et al. (2020a) show that neural networks can avoid the curse of dimensionality. Poggio et al. (2020b) show that neural networks have complexity control by analyzing the normalized gradient. Veit et al. (2016) find that ResNets behave like ensembles of shallow networks by examining the paths the data travels through.

**Algorithmic stability** Bousquet & Elisseeff (2000) use McDiarmid's method to show that algorithmic stability implies good generalization error. Hardt et al. (2016) analyze the algorithmic stability of SGD.

**Studies on the gradient** Sankararaman et al. (2020) study the impact of gradient confusion on the speed of neural network optimization concluding that SGD is fast when gradient confusion is low. Yin et al. (2018) Jain et al. (2018) study the effect of gradient diversity on the speed of optimization and show that greater gradient diversity leads to large mini-batches more effectively speeding up SGD. Fort et al. (2019) investigate the stiffness of neural networks and measure the cosine similarity between gradients as an approximation of how much one data point reduces loss on another. Chatterjee (2019) connects gradient averaging to algorithmic stability and generalization. Jacot et al. (2018) study generalization in the case that the network width reaches infinity, in which case dynamics are governed by the Neural Tangent Kernel. Arora et al. (2019) find that in practice, finite width CNTKs generalize better than infinite width models, indicating there may be some advantage to finite width. Novak et al. (2018) empirically study sensitivity measures such as the input output jacobian, and find that this correlates with generalization. Arpit et al. (2017) study memorization in neural networks. One quantity they consider is the gradient of the loss with respect to a particular input sample.

**Variance reduction** A key idea in traditional machine learning is that of variance reduction. Boosting and bagging are employed as techniques to reduce variance by averaging over different (uncorrelated) models.

**SGD as noisy GD** Mandt et al. (2016) show that SGD can be interpreted as an Ornstein-Uhlenbeck process. Another line of work has investigated the anisotropic noise of SGD as compared to GD. Since the SGD update samples a minibatch to calculate the gradient instead of using the whole training set, they view the SGD update as a noisy version of the GD update. Zhu et al. (2019) explore the effectiveness of the anisotropic noise in SGD at escaping bad minima of the loss function.

**Assessing Layerwise Convergence** Raghu et al. (2017) Morcos et al. (2018) use canonical correlation analysis to reveal insights into the convergence across different neural networks. They consider each neuron to be a vector, with each entry corresponding to the neuron value on a training data point, and each layer to be a subspace spanned by its neurons' vectors.

**Assessing Activation Patterns** Hanin & Rolnick (2019) investigated activation patterns in ReLU networks and found that they use surprisingly few such patterns. Cogswell et al. (2016) improved generalization of neural networks by introducing a regularizer that de-correlated activations.

**Dropout** Neural networks may be explicitly regularized using techniques such as dropout. Dropout imposes regularization by dropping nodes in a way that is typically sampled from a Bernoulli distribution. The resulting model can be considered an ensemble. Gao et al. (2019) investigate the effects of separating the forward from the backward pass.

## 3 ON FUNCTIONS

Mathematically, a function is a binary relation between two sets that associates each element of the first set with exactly one of the other set. ReLU functions clearly satisfy this definition, and all images pass through the same ReLU function. Yet, we will argue that each image is more accurately considered to have its own function, because the overall ReLU function is not particularly smooth, and does not, for example, satisfy the nice properties of polynomials use in function analysis. Although each image function is calculated on the same weights, we find that the varying ReLU paths make them sufficiently different.

## 4 GENERALIZATION

At the start of training, both the training data and test data are assumed to be drawn from some distribution $\mathcal{D}$. However, during training, input data are sampled uniformly from $\mathcal{T}$, the training set, call this distribution $\mathcal{U}$. The key question behind generalization is why optimizing over $\mathbb{E}_{\mathcal{U}}$ results in low loss over $\mathbb{E}_{\mathcal{D}}$. In this work, we show that neural networks learn different functions for the same image during training, and that this function diversity may contribute to generalization.

## 5 FULLY CONNECTED NETWORK GRADIENTS

We use $[N] := \{1, 2, ..., N\}$. SGD computes the following update on neural network parameters

$$w_{t+1} = w_t - \eta \nabla_{w_t} \mathcal{L} \tag{1}$$

where $\nabla_{w_t} \mathcal{L}$ denotes the gradient of the loss with respect to the weight parameters, and $\eta$ the learning rate. We focus on the case of minibatch SGD, where the gradient update is computed over a batch of data. For a particular node $n$ evaluated on an image $x_i$, it will have incoming backpropagated gradient at time $t$ of $z_i^{(t,n)}$. Let $N$ be the number of nodes in the network and $L$ the number of layers. We assume each node has an index, $n$, in $[N]$, and furthermore for the sake of notation that all nodes in the first layer are indexed $\{1, ...\ell_1\}$, that the second layer is indexed $\{\ell_1, ..., \ell_2\}$ etc. with $\ell_L = N$. We will also write $\{\ell_{-1}, ..., \ell_0\}$ to denote the coordinates of the input. Let $\overrightarrow{a_i}^{(t,\ell_j:\ell_k)}$ be the vector of activations for image $x_i$ at time $t$ from nodes in layers $j$ to $k$. We define a function $p_i^t : [N] \rightarrow \{0, 1\}$ that takes the index of the neural network's $N$ nodes, and returns 1 if it is on for image $i$ at time $t$ according to the ReLU activation function and returns 0 otherwise. With some abuse of notation, if we omit the function argument and write $p_i^t$, we define this as the vector of length $N$ containing $p_i^t(n)$ as each entry. Let $B_t$ be the minibatch sampled at time $t$. So the gradient update for a particular node $n$ in layer $\ell_k, k \in [L]$ at time $t$ is

$$\sum_{i \in B_t} z_i^{(t,n)} \cdot p_i^t(n) \cdot \overrightarrow{a_i}^{(t,\ell_{k-2}:\ell_{k-1})} \tag{2}$$

for the weight parameter and $\sum_{i \in B_t} z_i^{(t,n)}$ for the bias parameter.

Let

$$f_i^{(t,n)}(x) = \left( z_i^{(t,n)} \cdot p_i^t(n) \cdot \overrightarrow{a_i}^{(t,\ell_{k-2}:\ell_{k-1})} \right) \cdot x + z_i^{(t,n)} \tag{3}$$

Then the overall function a fully connected node $n$ computes at time $t$ for input $x$ is

$$f^{(t,n)}(x) = \left( \sum_t \sum_{i \in B_t} f_i^{(t,n)}(x) \right) \cdot p_x^t(n) \tag{4}$$

where $p_x^t(n)$ denotes the path function for image $x$ (not necessarily in the training set), and the layer activations are given by

$$f^{(t,1:\ell_1)}(x) := [f^{(t,1)}(x), f^{(t,2)}(x), ..., f^{(t,\ell_1)}(x)] \tag{5}$$

where the square braces denote concatenation and the layer function for $\ell_j$ may be defined recursively.

$$F^{(t,\ell_j)}(x) = f^{(t,\ell_{j-1}:\ell_j)}\left(\cdots f^{(t,1:\ell_1)}(x)\right) \tag{6}$$

We will define the history of the neural network as

$$\mathcal{H}_T = \{\sum_{i \in B_1} f_i^{(1,[N])}, ..., \sum_{i \in B_T} f_i^{(T,[N])}\} \tag{7}$$

The activations $\overrightarrow{a_i}^{(t,n)}$ for an input image $x_i$ are found by projecting the image onto the history of the network, e.g.

$$\overrightarrow{a_i}^{(t,1:\ell_1)} = f^{(t,1:\ell_1)}(x_i) \tag{8}$$

We will use the notation

$$p_i^t | \mathcal{H}_t \tag{9}$$

to project the path $p_i^t$ onto the history, i.e. to run the forward pass for $F^{(t,\ell_L)}$ fixing $p_x^t(n)$ in Equation 4 to be equal to $p_i^t$ for all $n$. Once enough updates have occurred, Equation 4 contains a term $f_i^{(t,n)}$ for all $i$ in the training set. It is therefore not obvious why continuing to update using only the $i$ in the training set does not yield a function that is very correlated on the training set and only produces low values of the loss on those points (and not on the test set). The gradient of neural networks is sometimes treated as a black-box in terms of generalization. However, we can exploit the fact that many neural networks use some form of dot product for each neuron to write the $a_i$ terms in Equation 4. Experimentally, we will measure the normalized dot products of these activations, as well as path overlap.

**Evaluating the same point using different functions** Let $t$ be the current time, $x_i \in B_t$ and $x_j \notin B_t$. At time $t$, data point $x_i$ is evaluated using Equation 6 which contains $f_j^{(t,n)}$ as in Equation 4. One main observation is that $f_j^{(t,n)}$ and $f_j^{(t-k,n)}$ for $1 \le k \le t-1$ may have different $p_x^t(n)$ terms in Equation 4 (i.e. $p_x^t(n) \cdot p_x^{t-k}(n)$ may not have high normalized dot product). That is to say, the same image at different times in training may follow a different evaluation path, i.e. the set of nodes activated for that image may be different.)

This also means that although each $f_i^{(t,n)}$ may be correlated with the *history* $\mathcal{H}_t$ of the network, and hence be correlated with the other *previous* $f_j^{(t-k,n)}$ for $1 \le k \le t-1$, the ReLU activation causes $p_j$ to be new enough that the network may not already correlated with $f_j^{(t,n)}$ at the current time $t$ (until $x_j$ next updates.) This encourages generalization, as the network, despite having seen $x_j$ before, does not pass it through a function too correlated with $x_j$. This means that although the network has seen image $x_j$ before, it learns it using a different function, encouraging generalization. In summary, we show experimentally that *Neural networks learn differing functions for the same image during training*

**Earlier functions are destroyed by subsequent ones** In an ensemble, typically an average of functions is taken to be the final predictor of the form $w_i g_i$. In the setup described above, although image $x_i$ may follow $p_i^t$ at time $t$, it may subsequently not be assigned that same path again. Hence, it cannot use all of $f_i^t$ in future rounds. We believe this also means that having extremely low correlation or independence is not as important as it is in ensembles. Rather, there is a tradeoff; more similarity may mean easier optimization, while somewhat less similarity may lead to more algorithmic stability, due to not re-using the same set of features.

**Which functions are optimized** We use $p_i^t | \mathcal{H}_t$ to be the function found projecting the path $p_i^t$ onto the history $\mathcal{H}_t$. Each data point $x_i$ at time $t$ will have loss $\mathcal{L}(p_i^t | \mathcal{H}_t)$ where $\mathcal{L}$ is some loss

function (e.g. the cross entropy loss). Suppose $x_i$ is in the updating batch at time $t$, i.e. $i \in B_t$, then $\nabla \mathcal{L}(p_i^t | \mathcal{H}_t)$ is directly calculated, and a gradient step is taken to descend on this loss function. Furthermore, since the function $p_i^t | \mathcal{H}_t$ depends on all previous images $x_j$ after the first epoch in training, it is not obvious why while descending on $\mathcal{L}(p_i^t | \mathcal{H}_t)$, the functions $\mathcal{L}(p_j^t | \mathcal{H}_t)$ are not also optimized for images $x_j$ in the training set only. We hypothesize that even if the network tries to optimize $\mathcal{L}(p_i^t | \mathcal{H}_t)$ in terms of prior $p_j^k | \mathcal{H}_k$ for $k < t$, this does not correspond to optimizing $x_j$ in terms of its current function $p_j^t | \mathcal{H}_t$.

From the last time $x_j$ updated, the network cannot re-run the forward pass on $x_j$, and $p_j^t$ will be reassigned according to the ReLU function and the current history $\mathcal{H}_t$. So the $i \in B_t$ do not have access to the end to end back-propagated gradient along the updated $p_j^t$ and instead calculate gradients along their own paths $p_i^t$ for $i \in B_t$. Each of these $p_i^t$ have some intersection with the previous $p_j^k$ for $k < t$, typically $p_i^t \cdot p_j^k > 0$. These paths $p_i^t$ serve to collectively update along $p_j^t$, but not by simple averaging or by linear combination of the whole paths. The sum of $f_i$ in Equation 4 do not correspond to a single back-propagated function, and are instead separate functions (Equation 3) stitched together according to the ReLU gate. When re-assigning $x_j$ to its new path, the $z_i^{(t,n)}$ in Equation 3 may not correspond to the evaluation path ahead in the network that $x_j$ will actually follow (because $p_j^t$ is unknown). The function $x_j$ follows at evaluation time is hence a sort of 'franken-function' which is constructed on the fly and is not explicitly optimized. It also does not correspond to simply adding together the $f_i$, as the $p_x$ term in Equation 4 must be applied. Additionally, the previous path that $x_j$ was following is likely to be destroyed; we find experimentally that the image does not follow $p_j^k$ at a later time $t$. Also, either all gradient updates to a node are applied (if it is on) or none are (if it is off), so data points may not selectively pick some functions to follow end-to-end.

Additionally, we find that training images $x_i$ and $x_j$ intersect in different places (along different paths) at different times in training, and that $x_i$ has highest dot product with different images throughout training. We interpret this to mean that *The ReLU gate encourages the use of different features at different times.*

We show experimentally that:

- For a particular previous $f_i^k$ $k < t$, each current updating $f_j^t$ have only partial path intersection with $f_i^k$. (They do not have complete access to all previous activations). Furthermore, they intersect on different nodes at different times. (Indicating they may learn from each other in different ways at different times)

- The $f_i^t$ at different times $t$ for some fixed image index $i$ are different. (Self similarity is not perfect.)

- The overall layer function, Equation 6 is not too smooth, supporting the case for considering $f_i^t$ for various $i$ as different functions.

- For a particular image i, the ranking of which functions $f_j^t$ are most path similar, for various times $t$, changes, indicating that which images are most similar to each other changes. This is in contrast, e.g., with K nearest neighbors, where the identity of the nearest neighbor remains fixed for fixed training data.

**CNN layers** the update for CNN networks may be computed comparably, with Equation 3 modified to contain a sum over patches $c_j$ of the original image

$$f_i^{(t,n)}(x) = \left( \sum_j z_i^{(t,n)} \cdot p_i^t(n) \cdot \overrightarrow{c_j}^{(t,\ell_{k-2}:\ell_{k-1})} \right) \cdot x + \sum_j z_j^{(t,n)} \tag{10}$$

## 6 EXPERIMENTS BETWEEN DATA POINTS

Equation 4 involves three terms; a $z_i$ term, an $a_i$ term (the activations of the layer before), and a path term $p_i$. In our experiments, we first focus on the $p_i$ term, as it reveals which nodes are on or off for a particular function. We use the usual definition of $p_i$ and allow the ReLU function on the current history $\mathcal{H}_t$ to determine if a node is on or off. Since the $a_i$ term in Equation 4 is dotted with $x$, which would be the activations of input $x$ at the layer before, we also graph dot products between activations at the same layer. We concatenate all activations into a single vector for a layer. We also conduct experiments on the whole function $f_i^t$ applied in isolation in Appendix Section C. Experimental details for training the model can be found in Appendix Section B.

### 6.1 ALEXNET CROSS PATH ALIGNMENT

We let $\odot$ denote the elementwise product. We omit the argument for $p_i^t$ and assume that it is $(\ell_{k-1} : \ell_k)$ We measure the quantity $\frac{(p_i^t \odot p_j^t) \cdot (p_i^{t-1} \odot p_j^{t-1})}{|(p_i^t \odot p_j^t)||(p_i^{t-1} \odot p_j^{t-1})|}$ , which determines the overlap between the intersection of paths $p_i^t$ and $p_j^t$ at the current epoch $t$ versus the previous epoch $t - 1$. This answers the question, does image $x_i$ always have access to the *same* set of activations from $x_j$? We see in Figure 1 that the answer is no. We hypothesize that this is the ReLU gate and SGD optimization encourages $x_i$ to use a different set of previous features from other images $x_j$, and this this diversity could encourage generalization.

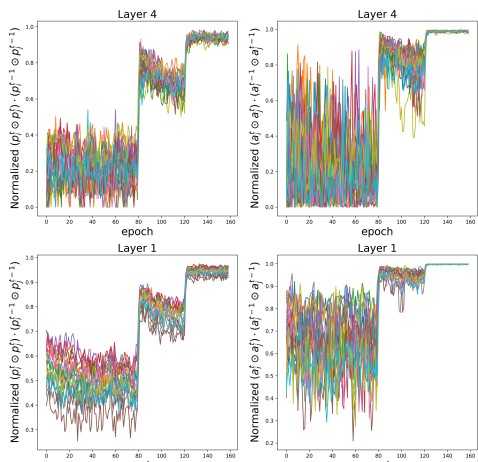

Figure 1: Cross path similarity $\frac{(p_i^t \odot p_j^t) \cdot (p_i^{t-1} \odot p_j^{t-1})}{|(p_i^t \odot p_j^t)||(p_i^{t-1} \odot p_j^{t-1})|}$ for AlexNet trained on CIFAR-10 data in the left column and the quantity $\frac{(a_i^t \odot a_j^t) \cdot (a_i^{t-1} \odot a_j^{t-1})}{|(a_i^t \odot a_j^t)||(a_i^{t-1} \odot a_j^{t-1})|}$ in the right column. The top row depicts the 4th layer of the network and the bottom row the 1rst layer of the network. We find that for a majority of training, there is not a perfect correlation for either quantity, indicating images intersect each other on different features at different points in training.

### 6.2 RESNET 18 ON CIFAR 10 CROSS PATH ALIGNMENT

For ResNet-18 trained on the CIFAR 10 dataset, we plot the normalized path overlap as we did for AlexNet. We plot our results in Figure 2. We find that in general, the correlations are higher than those found in AlexNet, but still not perfectly correlated until late in training. ResNet is also significantly deeper than AlexNet, and a small difference in each layer cumulatively leads to a different overall path. **VGG-19 on CIFAR 10 cross path similarity** We have similar findings for VGG-19 in Appendix Figure 16.

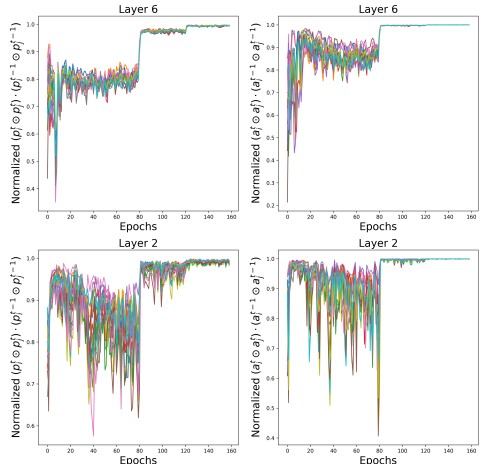

Figure 2: Cross path similarity $\frac{(p_i^t \odot p_j^t) \cdot (p_i^{t-1} \odot p_j^{t-1})}{|(p_i^t \odot p_j^t)||(p_i^{t-1} \odot p_j^{t-1})|}$ for ResNet-18 trained on CIFAR-10 data. Bottom row depicts layer 2, top row depicts layer 6.

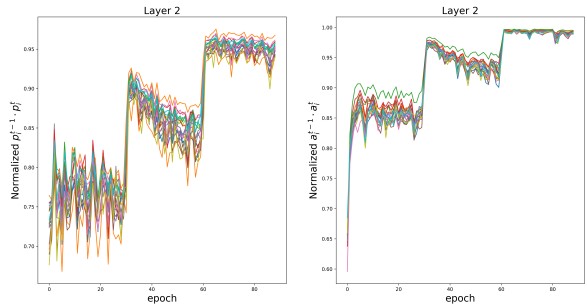

Figure 3: Cross path similarity $\frac{(p_i^t \odot p_j^t) \cdot (p_i^{t-1} \odot p_j^{t-1})}{|(p_i^t \odot p_j^t)||(p_i^{t-1} \odot p_j^{t-1})|}$ for ResNet-18 trained on ImageNet data.

### 6.3 RESNET 18 ON IMAGENET CROSS PATH ALIGNMENT

For ResNet trained on the ImageNet dataset, we plot the normalized distribution of $\frac{(p_i^t \odot p_j^t) \cdot (p_i^{t-1} \odot p_j^{t-1})}{|(p_i^t \odot p_j^t)||(p_i^{t-1} \odot p_j^{t-1})|}$ We plot our results in Figure 3. We find that this distribution is not perfectly correlated, though it becomes more correlated later in training.

### 6.4 ALEXNET CIFAR 10 PATH DISTANCES

We run experiments to asses how different the $f_i^t$ are for different $i$. We try to asses whether the overlap between path vectors is related to the distance between original points in data space. We plot our results in Figure 4. We find that there is not a clean linear relationship between distance in original space and overlap in path vectors, supporting the case for considering each image update as its own function, and again indicating that there is a kind of function diversity present in ReLU network training.

**Resnet-18 on CIFAR-10**: We have similar findings for Resnet -18 trained on CIFAR-10 in Appendix D

## 7 EXPERIMENTS ON SELF SIMILARITY

In this section, we seek to asses how different the paths and representations are for a **single** image throughout the course of training.

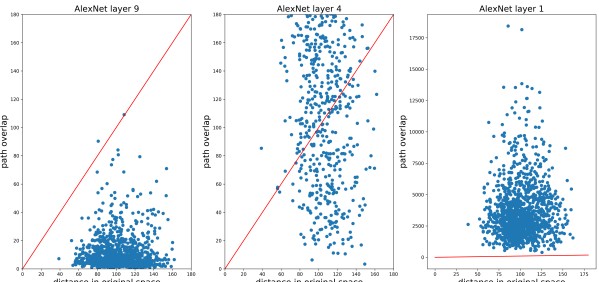

Figure 4: Path overlap $p_i^t \cdot p_j^t$ versus distance in original space $||x_i - x_j||_2$ for AlexNet trained on CIFAR 10. Layer 6 shown on the left and Layer 2 shown on the right. We find that there is not a clean linear relationship between distance in original space and overlap in path vectors, supporting the case for considering each image update as its own function.

### 7.1 ALEXNET ON CIFAR 10

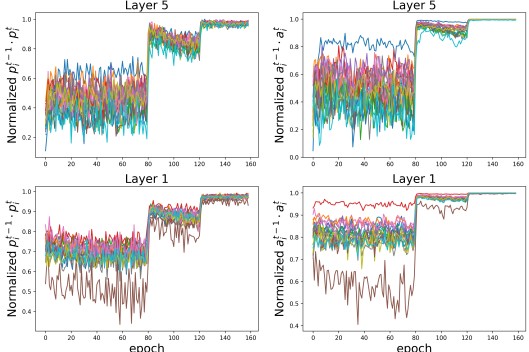

Figure 5: Self similarity $\frac{p_i^{t-1} \cdot p_i^t}{||p_i^{t-1}||||p_i^t||}$ for AlexNet trained on CIFAR 10 data. Bottom row depicts layer 1, top row depicts layer 4.

For AlexNet trained on the CIFAR 10 dataset, we plot the normalized distribution of $p_i^t([\ell_{k-1} : \ell_k]) \cdot p_i^{t-1}([\ell_{k-1} : \ell_k])$. We plot our results in Figure 5. We find that especially in the beginning of training, $p_i^t$ and $p_i^{t-1}$ are not perfectly correlated, indicating that the same image $x_i$ is passing through different nodes at different times $t$.

## 8 THE IDENTITY OF THE MOST SIMILAR IMAGE CHANGES

For three images we rank 128 other data points and restrict the resulting plot to the nearest 6 neighbors, where rank of 0 means it is the nearest neighbor. We use AlexNet on CIFAR 10 layer 2. We plot our results in Figure 6. **Result using activation similarity** We present analogous results to above using normalized activation similarity for layer 4 and 2 of AlexNet in Appendix E

### TAKEAWAY FROM EXPERIMENTS

We find that in all the architectures we examined, the self similarity and the intersection similarity between data points changed throughout (especially the early part) of training. These variations may lead to generalization for the similar reasons to ensembling over different models. Unlike in ensembling, we see that $f_i^t$ is destroyed at a later time $f_i^{t+k}$, because $x_i$ may not pass through the whole $p_i^t$ and instead may only use part of it (the part that intersects with $p_i^{t+k}$). Furthermore, each

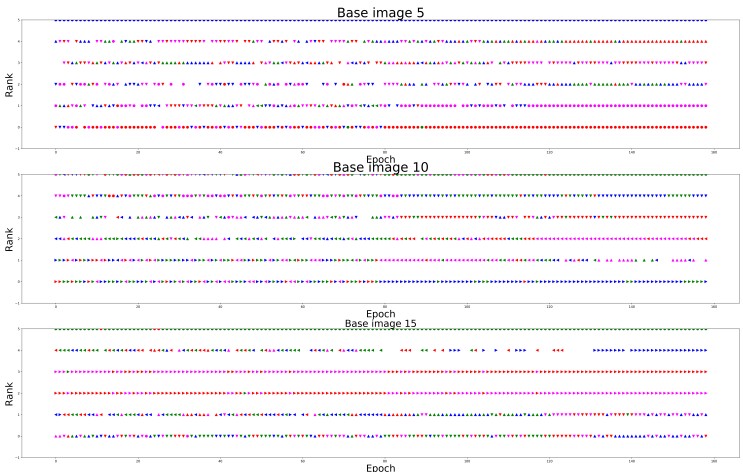

Figure 6: Plotting closeness to baseline image $x_j$ for $j = 5, 10, 15$. Closeness defined as path overlap $p_j^t \cdot p_i^t$ for other image i. AlexNet trained on CIFAR 10 data. We find that images $x_j$ can have different nearest neighbors at different times during training.

other image function $f_j^t$ is only correlated with *parts* of $f_i^t$ (on certain nodes.) This may explain why averaging is not necessary for variance reduction.

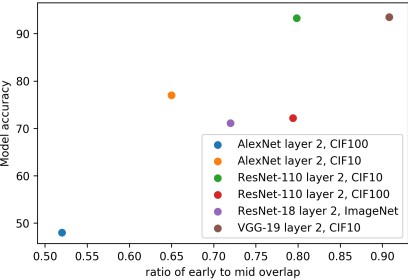

Figure 7: Early to mid cross path overlap ratio versus generalization.

**Ratio of early to mid cross path overlap** Finally, we can utilize the idea of cross path overlap in an attempt to predict generalization error. We hypothesize that higher (but not perfect) cross path overlap early is good, as typically the learning rate is high and this reflects a large number of shared features in the data On the other hand, high cross path overlap after decay may be bad, as diverse features are no longer being learnt. We plot (averaged over training data) the ratio of early (average over epochs before first decay) to mid (averaged over second stage of training) cross path overlap for various models against their generalization error in Figure 7. There is a roughly linear relationship, and improving such predictions could be an interesting avenue for future work.

## CONCLUSION

We have studied neural networks from the perspective of considering each image, updating at a particular time, as a different function. We have highlighted some properties of practical neural networks, namely that they learn varying representations for a single image, and that each image intersects each other image in different places during training. We hypothesize that this diversity of representation could lead to better generalization, by encouraging the model to learn the same image in different ways (akin to model ensembling) and by improving algorithmic stability.

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
