# OpenReview forum: "Image Functions In Neural Networks: A Perspective On Generalization"
_ICLR.cc/2022/Conference — ICLR 2022 Submitted_

### Official Review · Reviewer_fEgL · 2021-11-02

**Correctness:** 2
**Technical Novelty And Significance:** 1
**Empirical Novelty And Significance:** 1
**Recommendation:** 3
**Confidence:** 3

**Main Review:**

Weaknesses :
- Introduction is a bit disorganised, feels a bit too much like a list of independent concepts, and it is hard to extract the relation with the contribution of this paper
- The paper is poorly written, poorly structured, and the results are extremely difficult to parse, in great part due to imprecise language. Too many sentences are vague and need to be clarified, with words such as “image” used to mean very different things. A few examples : “indicating images intersect each other on different features at different points in training” ; “the ReLU gate and SGD optimization encourages xi to use a different set of previous features from other images xj” ; “supporting the case for considering each image update as its own function, and again indicating that there is a kind of function diversity present in ReLU network training” ; “the same image xi is passing through different nodes at different times t”
- Due to this lack of rigour, I struggled to understand the main message of the paper. This is not clear either from the experimental section : the fact that activations / paths become more and more correlated from one epoch to the next through training is not surprising,

Comments :
- “The key question behind generalization is why optimizing over E_U results in low loss over E_D” : I disagree with this. In typical studies of generalisation, the train and test data come from the same distribution (when they do not, the problem under study is out-of-distribution generalisation). The question is why optimisation over the empirical distribution (at finite sample size) leads to good generalisation over the full distribution.
- Fig. 4 : the overlaps need to be normalised on both axes for this plot to be meaningful
- Fig. 7. : The correlation which appears can hardly be named a “roughly linear relationship”

Nits :
- “Mathematically, a function is a binary relation between two sets that associates each element of the first set with exactly one of the other set” : not all functions are one-to-one mappings…
- “the second layer is indexed {l_1, ..., l_2}” : shouldn’t it start at l_{1+1} ?
- “We will also write {l_{-1}, ..., l_0} to denote the coordinates of the input” : is l_{-1} negative ? Shouldn’t l_0 =1 ?
- Eq. 2 : second dot product should be a simple multiplication, no ?

**Summary Of The Paper:**

This paper analyses how the representations of neural networks evolve during training, in particular which ReLU nodes are activated.
The authors attempt to show how this leads to a diverse set of functions being learnt.

**Summary Of The Review:**

Overall, I found this paper extremely vague and confusing, and struggled to follow the message the authors try to convey.
In the current state, I do not think this paper reaches the standards of a venue such as ICLR.

---

### Official Review · Reviewer_ec11 · 2021-11-02

**Correctness:** 2
**Technical Novelty And Significance:** 1
**Empirical Novelty And Significance:** 2
**Recommendation:** 3
**Confidence:** 2

**Main Review:**

This is an entirely empirical work. The authors look into the activations within a neural network (as either a path or a set of real-valued activations) and examine how such activations change over time during training for the same pair of images. What the authors basically show is that early during training, the activations change over time before they settle when the learning rate is sufficiently small. I do not think this is an insightful finding because this is what would be expected by "training" the network: we expect the weights and bias terms to change during training (hence activations would change) until the network is fully trained.

The authors' arguments about how such findings help understand generalization are heuristic. The authors claim that because the activations change over time, the network learns to use different features for the same image, and that improves generalization! Unfortunately, most of the claims in the paper are of this nature. For example, the authors state that "less similarity may lead to more algorithmic stability, due to not re-using the same set of features", where similarity here is the similarity of the path of activations. However, if we consider the simplest possible neural network, which is a logistic regression classifier, similarity will be high throughout training and it is hard to argue that logistic regression is less stable than deep neural networks (e.g. in the sense defined by  Bousquet & Elisseeff 2002). Unfortunately, such statements are the main contributions of the paper and they are neither justified nor precisely stated.

The paper also needs a lot of improvement in its presentation. Some issues include:
- The axis labels in the figures are too small to read.
- The line spacing in some paragraphs is different from the reset of the paper (e.g. Paragraph 2 in Page 1 and the paragraphs in Page 5).
- The related works section lists papers related to deep learning but I don't see the connection to the present work. In addition, there are lots of research papers that study activations/representations in the neural network that are not cited, which would be more relevant to the present paper.
- In Equation 4, the same symbol t is used both inside and outside the summation.


The paper contains many typos. Examples:
- Page 1: "PAC Bayes based bounds Dziugaite & Roy (2017) approaches exploring" --> "PAC Bayes based bounds Dziugaite & Roy (2017) and approaches exploring"
- Page 2: "using PAC-Bayes He & Su (2019)" --> "using PAC-Bayes. He & Su (2019)"
- Page 2: "jacobian" --> "Jacobian"
- Page 4: "the network may not already correlated" --> "the network may not already be correlated"
- Page 4: "that Neural networks" --> "that neural networks"
- Page 4: "learn differing functions" --> "learn different functions"
- Page 9: "features in the data On the other hand" --> "features in the data. On the other hand"


**Summary Of The Paper:**

The authors present empirical results about the correlation between the activations in a neural network across time for a fixed pair of images.  They show that for the same pair of images xi and xj, the activations change over time until they settle when the learning rate is sufficiently small. The authors claim that this provides insight into why neural networks generalize, by arguing that the network uses different features at different epochs during training.

**Summary Of The Review:**

The empirical findings reported in the paper are not insightful and the paper makes claims that are heuristic and not justified.

---

### Official Review · Reviewer_M7Dz · 2021-11-03

**Correctness:** 2
**Technical Novelty And Significance:** 2
**Empirical Novelty And Significance:** 3
**Recommendation:** 3
**Confidence:** 4

**Main Review:**

The paper analyze the dynamic of “image functions” and claims that analyzing the dynamic of image functions could potentially explain the generalization. I have a few questions and concerns.

1. The major claim of the paper is that neural networks generalize since they learn different functions for the same image during training (page 4). This statement is non-trivial and I cannot see direct link between generalization and the dynamics. When the paper talks about generalization, does it mean the generalization error is small? If this is the case, the paper requires some formal mathematics to bridge the gap between the dynamics of image functions and generalization error.

2. Why call it image functions? It seems that the framework is applicable for any fully-connected networks. The experiment section does not explain this neither. The only thing related to images is CIFAR10 and ImageNet. Based on the naming, I would expect to see more connection to low-level vision function or filters.

3. Another major finding of the paper is that the function for images would be different in different time step. This statement seems to be expected as the network parameter is keep updating via SGD. In particular, the function represented by neural networks is expected to change over time, it would be great if the authors can provide more clarification about this.

4. I found the notation is a bit hard to follow. Section 5 introduces various notions such as path, history, image projection, it would be better to provide a high level idea at the beginning of the section. For me, it seems that section 5 is writing the explicit gradient formulation for fully connected networks.

5. As the major claim is about generalization, it is better to theoretically or empirically provide some evidence related to generalization error. Although figure 7 provide some hints, but it is still elusive why this improves the generalization.

6. The abstract states two way to improve generalization. However, it seems that the claim is not explicitly examined in the experiment section. For instance, how to concretely encourage the model to learn the same image in different ways?

Overall, the paper addresses an important problem. The idea could be potentially interesting, but I found the writing and presentation is a bit confusing.

**Summary Of The Paper:**

The paper analyze the dynamic of “image functions” and show that analyzing the dynamic of image functions can explain the generalization. In particular, the authors visualize the cross path similarity and overlap to provide potential insights to generalization of neural networks.

**Summary Of The Review:**

The idea could be potentially interesting, but I found the writing and presentation is a bit confusing.

---

### Official Review · Reviewer_ypTe · 2021-11-03

**Correctness:** 3
**Technical Novelty And Significance:** 2
**Empirical Novelty And Significance:** 2
**Recommendation:** 3
**Confidence:** 3

**Main Review:**

The primary strength of the paper is novelty in that it could be a contribution of explaining why neural networks generalize despite violating traditional ML theory and rules of thumb.  The idea of "image functions" seems different from other approaches to explaining generalization.  Also, although the results are presented for the specific case of classification on images with ReLU networks trained with SGD, it seems that the ideas could be more general in future follow-up work.

However, there are some weaknesses with the paper that point to more work and clarity before publication.  The main application shown is in Figure 7 where a candidate statistics namely ratio between early and mid cross path is correlated with generalization performance.  But there are only a few points for comparison which makes it difficult to be confident in the claim that there is a good correlation between the statistic and generalization.  For example both VGG-19 and ResNet-110 perform about the same for CIFAR10, but have significantly different statistics.  For the other data sets such as CIFAR100 there are only two points to establish the correlation.  In general, it would be better to separate the experiments by data set and have more comprehensive results such as showing more statistics in other layers, etc.

Also, the results show specific statistics for specific layers, but how was the layer chosen?  Do all layers show the same trend or is there a criterion to choose which layer matters?

Another thing that would help the reader would be to relate the proposed approach to the previous approaches.  Although the contribution is mostly experimental, still it would be useful to know whether the implication is variance reduction such as in ensemble methods or is it some other theoretical idea that the results point to.  Adding some discussion on this in the prior work section and/or elsewhere would help connect the ideas in the paper to what is known in the literature.  Various parts of the paper hint at algorithmic stability, but it is difficult for the reader to see the strong connection here.

Finally there are some minor typos, etc. that can be improved.  The claims made right before Section 2 is somewhat difficult for the reader to interpret at that point as the notation hasn't been explained yet before being used.  Similarly Section 3 is also a bit cryptic before reading the later sections which describe the actual details.  The description of the details in Section 5 may benefit from a diagram and also there may be typos involving the specification of the node/layer indices such as l_1 being repeated for both the last node in layer 1 and the first node in layer 2.  Overall, the explanation of what exactly is an image function and the definition of F could be clarified by perhaps showing more concrete examples, exactly because this idea is somewhat novel.  Figure 4 and 5 captions also seem to have typos for which layer is depicted.

**Summary Of The Paper:**

This paper proposes a different look at why neural networks generalize despite optimizing to zero training error, over-parameterization, etc.  The contribution is mostly experimental in the sense of computing various statistics of a model during training and correlating those statistics with generalization performance.  The core idea is to define "image functions" which are determined by the training image and the current training iteration.  Correlation statistics on these functions for different and same training images show patterns in the training dynamics, which may indicate generalization for example as training continues towards zero error the statistics still vary.

The paper focuses on classification with CIFAR10 and ImageNet with AlexNet, VGG, and ResNet models.  In particular the results are specialized for ReLU networks training with SGD.

**Summary Of The Review:**

Overall, the paper tackles an important problem, namely explaining generalization via an experimental study and using a novel idea of "image functions".  The results indicate promise, however the fact that the experiments do not provide a comprehensive view such as showing the statistics for all or many layers and having more points to establish the correlation in Figure 7 make the paper a little premature.  Also the description of the details is somewhat unclear, and more importantly there isn't a good discussion of how this approach relates to previous work.  It is claimed that the evidence shows algorithmic stability but this isn't justified directly, and the reader also isn't clear whether the results show variance reduction or some other idea that leads to generalization.

---

### Decision · Program_Chairs · 2022-01-20

**Decision:**

Reject

**Comment:**

In an attempt to understand generalization, this paper aims at understanding the dynamics of functions presented by the network for different images in the training set. Authors look at activation patterns (whether a ReLU activation is on or off) as a way of characterizing the active paths in the network and approximating the function presented by the network for each image. Authors study different related statics (eg. correlation) and how they evolve during training including.

Pros:
- Understanding the dynamics of training, how diversity is encouraged by the training procedure and its relationship to generalization is an important problem.
- This paper takes an empirical approach and tries to make interesting empirical observations about the dynamics of the training.

Cons:
- The paper is poorly written in terms of structure, making clear arguments with enough evidence, notation, etc.
- Some empirical trends are shown but their connections to the main claim of the paper about generalization is very weak. The main attempt to connect the observations to generalization is Fig. 7 which shows model accuracy correlated with the ratio of early to mid overlap. This is problematic both because it only has 6 data points and also because a simple correlation analysis is not enough to establish this claim which is more about the cause of generalization.

Reviewers have pointed to various concerns including but not limited to clarity of the paper, lack of rigorous arguments, not providing enough evidence for the arguments, etc. Unfortunately, authors did not participate in the discussion period.

Given the above concerns, I recommend rejecting the paper.